# Composition and Structure of *Arabidopsis thaliana* Extrachromosomal Circular DNAs Revealed by Nanopore Sequencing

**DOI:** 10.3390/plants12112178

**Published:** 2023-05-30

**Authors:** Pavel Merkulov, Ekaterina Egorova, Ilya Kirov

**Affiliations:** 1Moscow Institute of Physics and Technology, 141701 Dolgoprudny, Russia; paulmerkulov97@gmail.com; 2All-Russia Research Institute of Agricultural Biotechnology, 127550 Moscow, Russia; egorova.ekaterina20021120@gmail.com

**Keywords:** extrachromosomal circular DNAs, *Arabidopsis*, long-read sequencing, transposable elements

## Abstract

Extrachromosomal circular DNAs (eccDNAs) are enigmatic DNA molecules that have been detected in a range of organisms. In plants, eccDNAs have various genomic origins and may be derived from transposable elements. The structures of individual eccDNA molecules and their dynamics in response to stress are poorly understood. In this study, we showed that nanopore sequencing is a useful tool for the detection and structural analysis of eccDNA molecules. Applying nanopore sequencing to the eccDNA molecules of epigenetically stressed *Arabidopsis* plants grown under various stress treatments (heat, abscisic acid, and flagellin), we showed that TE-derived eccDNA quantity and structure vary dramatically between individual TEs. Epigenetic stress alone did not cause eccDNA up-regulation, whereas its combination with heat stress triggered the generation of full-length and various truncated eccDNAs of the *ONSEN* element. We showed that the ratio between full-length and truncated eccDNAs is TE- and condition-dependent. Our work paves the way for further elucidation of the structural features of eccDNAs and their connections with various biological processes, such as eccDNA transcription and eccDNA-mediated TE silencing.

## 1. Introduction

Extrachromosomal circular DNAs (eccDNAs) are a type of double-stranded DNA that has been found in the cells of a variety of organisms, including humans, animals, and plants. The first report regarding DNA circularization was made in 1962, followed by Hotta and Basel’s experiments, in which eccDNAs were discovered in boar sperm using electron microscopy [1]. Their results provided the first indication of the occurrence of eukaryotic DNA in a circular form. Circular DNAs were initially considered to originate from the mitochondria, as they were co-isolated with organellar DNA [2]. After examining mutant yeast lacking mitochondria, it was discovered that circular DNAs have nuclear origin [3]. Using electron microscopy, the first plant nuclei-derived small circular DNA forms were investigated in wheat and tobacco (Kinoshita et al., 1985). Two-dimensional (2D) gel electrophoresis was used to distinguish eccDNA structural forms in different families of higher plants (*Asteraceae*, *Brassicaceae*, *Fabaceae*, and *Poecea*). Finally, the advent of high-throughput sequencing and bioinformatic methods facilitated insight into the diversity of circular molecules that comprise a plant circulome [4,5]. Genome-wide analysis of eccDNA-producing loci has been carried out for several plant species [5,6,7,8,9,10,11]. These studies indicated that different genomic loci can produce eccDNAs in plants, including genic and intergenic regions. A major source of eccDNAs in plant genomes is transposable elements (TEs) [4]. It was proposed that TE-derived eccDNAs are generated through homologous recombination and nonhomologous end joining of linear reverse-transcribed DNA of LTR retrotransposons [8,12]. Based on this, using eccDNAs as a marker for TE mobility was proposed. An early method of plant eccDNA sequencing, known as Mobilome-seq, was published by Lanciano et al. [6]. In brief, this approach employs exonuclease to remove linear DNA, followed by Phi29 polymerase random rolling-circle amplification of intact circle molecules and Illumina short-read sequencing. After read mapping the genome, further data processing is often focused on the detection of split-reads, including eccDNA borders. Wang et al. [11] used Mobilome-seq to conduct an extensive investigation of the eccDNA landscape in several *A. thaliana* tissues. The results of Lanciano et al. in [6] and Wang et al. [11] demonstrated the diversity and high organ/tissue specificity of eccDNAs. Applying the Mobilome-seq approach facilitated the detection of potential transposition activity for a number of TEs, including *EVD* and *ONSEN* in *A. thaliana*; Tos17, PopRice, *Houba* in rice; *Nightshade* in potato; and *Alex1* and *Alex3* in carrot [5,6,8,10,13]. 

Rolling-circle amplification (RCA) products have a concatemer structure with several tandemly organized monomers that correspond to full eccDNA molecules. Long-read sequencing of RCA products is particularly appealing for eccDNA de novo identification due to this property. This led to the development of the long-read eccDNA sequencing method known as CIDER-seq [14,15]. This technique has been used to profile eccDNAs and full-length viral DNA in a variety of species, including plants [14,15]. Size-based selection of extrachromosomal DNA (rather than exonuclease-assisted linear DNA removal) and sequencing of obtained RCA amplicons using PacBio SMRT are the core components of the CIDER-seq technique. PacBio sequencing of *A. thaliana* eccDNAs revealed a significant percentage of TE-originated eccDNAs, primarily *Helitron*, *DNA/MuDr* DNA transposons, and *Gypsy/Copia* LTR-retrotransposons. This is an additional compelling proof of TEs’ enormous influence on plant circulome development. Unfortunately, the long-read sequencing approach has not been extensively used for eccDNA sequencing, leaving structural features of plant eccDNAs unexplored. 

The goal of this study was to learn more about the structure and composition of eccDNAs found in *Arabidopsis thaliana* plants grown under TE-relaxed circumstances (epigenetic stress) in combination with various treatments (ABA, heat stress, and flagellin). Using long-read nanopore sequencing, we dissected several TE- and gene-derived eccDNAs and showed that a single TE can produce eccDNAs that vary in structure and composition. We proved that under stress, wild-type (Col-0) and *ddm1* (decrease in DNA methylation 1) mutant plants’ eccDNA structures could differ dramatically. Finally, our work demonstrates that nanopore sequencing is a useful tool for the rapid identification of active TEs and illuminating circulome changes in plants.

## 2. Results

### 2.1. Extrachromosomal Circular DNA (eccDNA) Sequencing of ddm1 Plants

Nanopore sequencing has rarely been used to analyze plant eccDNAs. To verify this method, we first sequenced the eccDNAs of the ddm1 Arabidopsis thaliana mutant, which has a high level of transposon activity and a well-known circulome composition [6,10]. Exonuclease-mediated linear DNA removal and rolling-circle amplification were used to enrich eccDNAs in a DNA sample [6]. The RCA products were then digested using a T7 endonuclease to debranch them (Figure 1A). Sequencing of ddm1 eccDNA via MinION Nanopore produced 257,563 reads. Col-0 (332,457 reads, 20× genome coverage) and ddm1 (194,354 reads, 25× genome coverage) whole-genome Oxford Nanopore Technologies (ONT) sequencing (WGS) data were utilized as a baseline to identify eccDNA-generating loci. We employed ONT reads containing two or more tandemly organized monomer units (sequence of one circle of the eccDNA molecule) and lengths > 500 bp to filter out reads that did not correspond to eccDNA RCA products. After this screening, 45,277 (17.6%) ONT reads remained for additional examination. Notably, for Col-0 (32,138 reads, or 9.7%) and ddm1 (21,279 reads, or 10.9%) WGS samples, the proportion of ONT reads with two or more monomers was considerably lower (the chi-square statistic using the Yates correction *p*-value was 0.00001). The TAIR10 genome was used to map the WGS and eccDNA concatemer reads, and the number of mapped reads for 500 bp genome windows was determined.

For both WGS samples, peaks of high coverage associated with tandemly organized repeats were discovered, including centromeric repeats and rDNA loci (Appendix A). These regions had significantly lower peak heights in the eccDNA ddm1 sample, indicating that our protocol sufficiently removed linear DNA from the sample. We evaluated the log2 ratio between the read counts for the eccDNA ddm1 sample relative to the WGS ddm1 to identify loci with high coverage by eccDNA reads. Four eccDNA peaks with log2 ratios greater than three were found (Appendix A); two of them, SRC2 (AT1G09070.1, Appendix A) and MKK9 (AT1G73500, Appendix A), were peaks on distal regions of chromosome 1. 

Three copies of the LTR retrotransposons EVD1 (AT1TE41575), EVD2 (AT1TE41580), and EVD5 (AT5TE20395) from the ATCOPIA93 family made up the other two peaks (Figure 1B). In contrast to EVD1 and EVD2, which have not been observed to produce eccDNAs, EVD5 is a well-known characteristic of ddm1 and epiRIL plants [6,16]. Additionally, EVD1 and EVD2 retrotransposons’ transposition activity was not reported [16]. Indeed, when we manually verified the alignment of eccDNA readings for EVD1 and EVD2, we found many SNPs that unmistakably showed that the reads originated from EVD5. The concatemer reads of EVD5 were incorrectly mapped to EVD1 and EVD2 because they are tandemly organized and share many similarities with EVD5.

Thus, nanopore sequencing of eccDNA-enriched DNA samples from A. thaliana is an effective method for detecting eccDNA-produced loci such as TEs and genes. Our findings show that EVD5 is the most active eccDNAs-producing TE in the ddm1 genetic background, which correlates with a prior discovery using short-read sequencing [6].

### 2.2. Epigenetic Stress Does Not Shift eccDNA Composition in A. thaliana

As TEs are the primary source of eccDNAs, we wondered if epigenetic stress could influence eccDNA formation. To test this, we cultured Col-0 plants in MS medium (ZA media) containing zebularine, a chemical demethylation agent, and α-amanitin, a PolII inhibitor, for 14 days. These toxins have the ability to reduce DNA methylation, hence allowing TE activation. Confirming the previous observations, Col-0 plants grown on toxin-supplemented media demonstrated a reduction in growth and root development (Appendix A). We sequenced eccDNA samples from Col-0 plants growing on control (K, no toxins) and ZA media. Total ONT readings per duplicate ranged from 138,861 to 236,853. We examined the log2 ratios of concatemer reads mapped to 500 bp windows between two biological replicates of ZA samples to determine a threshold. We discovered that 99.6% of the genomic windows had a −log2 ratio of two. Based on this, we established a log2 threshold of three and a minimum length of 1000 bp for loci expressing eccDNAs. Additionally, only loci found in both biological replicates were chosen. We analyzed the number of reads per 100,000 reads in K and ZA variants using these criteria and determined the log2 ratio of ZA to K (log(ZA/K)). Surprisingly, we found only one locus (Chr3: 13563500…13564500) that produces eccDNAs (Appendix A). This region overlaps with the AT3TE55175 transposon. Although the data indicate that epigenetic stress alone is insufficient to increase eccDNA production, we anticipate that some loci may still produce a tiny amount of eccDNAs that are below the current detection limit.

### 2.3. Detection of eccDNAs under Abiotic, Hormone, and Flagellin Treatment of Epigenetically Stressed Plants

Previous investigations claimed that plants grown on ZA media and subjected to stress, such as heat stress, are stimulated to produce eccDNAs [10,17]. We combined epigenetic (ZA) stress with each of the following stimuli to detect stress-responsive eccDNA loci: heat stress (HS), flagellin treatment (Flg), and abscisic acid treatment (ABA). EccDNA sequencing of these three variations yielded 306,796, 399,064, and 681,418 reads, respectively, in two replicates. Concatemer reads with two or more monomer units were then chosen and aligned to the TAIR10 genome. We used Fisher’s exact test with multiple corrections (Benjamini–Hochberg correction for multiple comparisons) to determine statistically significant peaks with increased read coverage in stressed samples compared with ZA samples. This test used only primary read alignments as input.

This analysis found no Flg- or ABA-responsive eccDNA peaks shared by two replicates (Appendix A). This could imply that no eccDNA-produced genomic loci were activated by these conditions, or that eccDNA production was low and the ONT reads were insufficient to detect them. Following that, we examined ONT data for HS samples and discovered seven eccDNA-producing loci that had considerably higher coverage by concatemer reads in HS ZA samples than in ZA samples (Figure 2A). These peaks corresponded to seven copies of the ATCOPIA78 family’s ONSEN LTR-retrotransposon: AT1TE59755 (ONSEN5), AT1TE71045 (ONSEN4), AT1TE12295 (ONSEN1), AT1TE24850 (ONSEN7), AT3TE89830 (ONSEN6), AT3TE92525 (ONSEN2), and AT5TE15240 (ONSEN3). ONSEN copies contributed differently to eccDNA contents; ONSEN3, ONSEN5, and ONSEN1 accounted for 92% of ONT reads, supporting previous results [18].

Thus, the data show that the eccDNA composition stabilized under ZA, ZA + ABA, and ZA + Flg treatments, whereas only heat stress significantly promoted eccDNA formation, but only from the ONSEN LTR-retrotransposon family.

### 2.4. ONT Reads Revealed Structure of eccDNA Molecules

We sought to shed light on the structure of eccDNA molecules generated from EVD and ONSEN TEs using long-read sequencing data. We created an original pipeline (eccStructONT) that made it possible to reconstruct eccDNA sequences. We performed ONT sequencing of the eccDNA molecules of ddm1 plants grown in vitro and exposed to mild heat stress (30 °C) to estimate the structure of ONSEN and EVD eccDNAs in the same conditions. When combined with in vitro cultivation, mild heat stress could activate ONSEN while ensuring ddm1 plant survival. To reduce noise, we only selected monomers that were repeated at least three times in the RCA concatemer read. We started by examining eccDNA molecules produced by EVD in ddm1 mutants. For EVD (Appendix A) and ONSEN elements (Figure 3A), the distribution of eccDNA lengths showed two peaks corresponding to full-length (fl_eccDNAs, near 5000 bp) and truncated (tr_eccDNAs, 1000 bp) eccDNAs. Next, we categorized various eccDNA molecules based on their region of origin in a TE frame. As seen in Figure 3B, the majority of EVD eccDNAs were fl_eccDNAs. 

The visualisation of separate eccDNA structure groups for ONSEN components suggests that fl_eccDNAs and tr_eccDNAs were the two major eccDNA forms (Figure 3C). ONSEN tr_eccDNAs are primarily derived from one or two LTRs. Accordingly, EVD and ONSEN elements differed significantly in terms of their eccDNA set composition, with ONSEN having a significantly greater proportion of LTR-derived tr_eccDNAs than EVD did. 

Next, we examined whether the ONSENs’ fl_eccDNAs to tr_eccDNAs ratio remained the same in wild-type plants under ZA HS stress as in ddm1 HS plants. Surprisingly, the structural study showed that all six ONSENs created primarily tr_eccDNAs, with very little fl_eccDNA production (Figure 4A). Again, LTR and LTR-adjusted internal ONSEN regions were the main sources of tr_eccDNAs (Figure 4A). Two primer pairs positioned on internal and LTR-adjusted parts of ONSENs were used in an RT-qPCR investigation to support these findings. The RT-qPCR results were corroborated using bioinformatic analysis (Figure 4B). 

The acquired data indicated that the eccDNA composition of a single TE can significantly differ between genetic backgrounds. As a result, it would be intriguing to investigate the genetic and environmental factors that influence eccDNA biogenesis.

## 3. Discussion

Genetic and environmental factors involved in eccDNA biogenesis in plants are not yet understood. However, it is clear that eccDNA composition and expression are dynamic processes. Previously, the results of a comparative analysis of eccDNAs in different *Arabidopsis* organs demonstrated that the eccDNA repertoire is influenced by unique routes and mechanisms involved in eccDNA production in different cell types [11]. Here, we did not find up-regulated eccDNAs under ZA Flg-stressed and ZA ABA-stressed conditions. It is possible that these stresses negatively regulated the expression of some TEs, as well as genes involved in the TE life cycle. The inhibition of TE-derived eccDNA production by stress was demonstrated in *Solanum tuberosum* and *S. commersonii* species [13]. Transposon *Nightshade* actively generates eccDNAs in control conditions, whereas cold stress inhibits eccDNA production [13]. This could explain why we did not detect any eccDNA peaks after applying stresses. Notably, the *Arabidopsis* genome has tens of TE families possessing transpositionally active elements [19,20]. However, the transposition of TEs from just a few families (e.g., *ATCOPIA93* (*EVD*), *ATCOPIA78* (*ONSENs*), *VANDAL*, etc.) has been observed in laboratory conditions. This implies that we still have very limited knowledge about conditions that can induce TE expression and transposition activity. New tools and techniques to trace TE transposition in different cell types and stress conditions are required.

Long-read sequencing followed by concatemer identification in raw ONT reads facilitated deciphering sequences of full-length eccDNAs, paving the way for the elucidation of eccDNA structure. Taking this into account, we investigated the structure of *EVD* and six *ONSEN* elements in the same genetic background (*ddm1*) and stress conditions (mild heat stress of in vitro growing plants). Our results indicated the presence of eccDNAs corresponding to full-length TE sequences (fl_eccDNAs) and certain TEs parts, including LTRs and LTR-adjusted regions (tr_eccDNAs). The ratios between fl_eccDNAs and tr_eccDNAs differed between *EVD* and *ONSEN* elements in *ddm1* under HS. In addition, fl_eccDNAs of *ONSEN* elements were almost not detectable in Col-0 plants under ZA HS stress; only tr_eccDNAs of different groups were identified. This further implies that the eccDNA repertoire may vary dramatically between individual TEs, stress conditions, and genetic backgrounds. The presence of different types of TE-derived tr_eccDNAs in a cell raised questions about the molecular mechanisms and possible functions of these molecules. A direct answer is that tr_eccDNAs are byproducts of intra- and intermolecular recombination events leading to fl_eccDNA conversion into tr_eccDNAs [21,22,23]. This scenario may have evolved as a defense mechanism against new TE integration. Most eccDNAs contain full-length LTR sequences that may function as promoters that trigger transcription from eccDNAs. It is well known that circular DNAs in the form of plasmids carrying target genes under promoter (e.g. 35S) are well transcribed, being transfected into the protoplast cells [24]. Moreover, RNA transcribed from eccDNAs was detected in humans [25] and plants [26]. Based on this, it could be proposed that RNA molecules transcribed from TE-derived eccDNAs may become substrates for RdDM-pathway-triggering TE silencing. 

## 4. Materials and Methods

### 4.1. Plant Material and In Vitro Growth Conditions

Seeds of *ddm1* (*ddm1-2*, F7 generation) were kindly provided by Vincent Colot (Institut de Biologie de l’Ecole Normale Supérieure (IBENS), Paris, France). *Arabidopsis* seeds were surface-sterilized with 75% ethanol (2 min) and washed with 5% sodium hypochlorite (5 min). After this, the seeds were rinsed with sterile distilled water 3 times. A total of 0.1% agarose was added to each tube and the seeds were resuspended and dripped on Petri dishes with Murashige and Skoog medium, supplemented with 3% of sucrose (PanReac AppliChem, Darmstad, Germany) and 1% of agar (PanReac AppliChem, Darmstad, Germany). Plates with seeds were sealed with Parafilm (Pechiney Plastic Packaging Company, Chicago, IL, USA) and kept in the dark at 4 °C for 3 days for vernalization and synchronous germination. Afterward, dishes were transferred into a light chamber with 16 h day/8 h night photoperiods for further growth.

### 4.2. Stress Conditions

Surface-sterilized Col-0 seeds were resuspended in 0,1% agarose and transferred to solid ½ MS medium supplemented with sterile filtered 4.6 ug/mL α-amanitin (Sigma-Aldrich, CAS 23109-05-9) and 9 ug/mL zebularine (Sigma-Aldrich; CAS 3690-10-6). For flagellin treatment, 2-week-old plants grown on ½ MS media supplemented with α-amanitin and zebularine were transferred to the equal ZA medium but also supplemented with 0.5 ug/mL flagellin (Flagellin from *Salmonella typhimurium*, Sigma-Aldrich, SRP8029-10UG). For ABA treatment, 2-week-old plants grown on ½ MS media supplemented with α-amanitin and zebularine were transferred to the equal ZA medium but also supplemented with abscisic acid at final concentrations of 100 uM. For heat-stress treatment, 2-week-old plants grown on ½ MS media supplemented with α-amanitin and zebularine were subjected to elevated temperature (4 °C for 24 h followed by 37 °C for 24 h, HS). For mild heat stress treatment 2 weeks-old Col-0 *ddm1* plants were subjected to 30 °C for 24 h. 

### 4.3. DNA Isolation

Total DNA was extracted by CTAB protocol [27] from plants immediately after stress (0 h recovery time). Briefly, 50 mg of young in vitro—grown *A. thaliana* plant tissues were ground with a pestle in a mortar with liquid nitrogen. A total of 0.5 mL of preheated to 75 °C CTAB1 buffer containing 6% of β-mercaptoethanol and 0.5% polyvinylpyrrolidone were immediately added to the frozen powder. A lysate was transferred to a 1.5 mL tube and incubated at 75 °C for 1 h. After cooling, an equal volume of chloroform was added to the sample. An upper water phase was transferred to a 1.5 mL tube containing 2 volumes of CTAB2 buffer. An obtained pellet was resuspended in 0.2 mL 1M NaCl, and DNA was precipitated with equal isopropanol volume followed by centrifugation. Pellet was washed with 70% ethanol and resuspended in nuclease-free water. An RNAse treatment was performed followed by isopropanol reprecipitation and ethanol washing. The obtained DNA was resuspended in nuclease-free water again for downstream analysis.

### 4.4. Extrachromosomal Circular DNA (eccDNA) Enrichment

For eccDNA enrichment, the protocol of Lanciano et al. [6] was used with some modifications. Briefly, linear DNA elimination was performed with PlasmidSafe DNase (LGC Biosearch Technologies, UK). For this, 750 ng of total DNA was combined with 2 μL ATP 25 mM, 5 μL 10× PlasmidSafe buffer, and 1 μL PlasmidSafe DNase (10 units) in 50 μL reaction volume. Incubation mode was 72 h at 37 °C with the addition of an extra amount of reagents (2 μL 25 mM ATP, 0.3 μL PlasmidSafe buffer, 1 μL PlasmidSafe DNase (10 units) after the first 24 h. Enzyme inactivation was performed at 70 °C for 30 min. The remaining DNA was precipitated by overnight incubation with 1/10V 3M sodium acetate (pH 5.2) and 2.5V absolute ethanol followed by centrifugation at 12,000× *g* for 30 min. The DNA pellet was subjected to random RCA reaction using illustra TempliPhi 100 Amplification Kit (GE Healthcare, catalog #25-6400-10). The reaction was carried out by adding 5 μL of TempliPhi sample buffer (the buffer was preliminary preheated at 95 °C for 3 min and then cooled to RT) to the DNA pellet. After adding 5 μL of the premix solution (5 μL Templiphi Reaction buffer, 0.2 Temliphi Enzyme mix) to the sample, the reaction was incubated at 28 °C for 72 h. After enzyme inactivation at 65 °C for 10 min, the sample volume was adjusted to 50 μL and purified with AmpureXP at 0.5× ratios according to the manufacturer’s instructions. A purified RCA product was debranched to remove the hyperbranched structures generated. The following reaction mix was prepared for debranching: 500 ng of RCA product, 5 μL of 10× reaction buffer, and 1 μL of T7 endonuclease I (New England Biolabs, M0302S) in 50 μL reaction volume. After the incubation at 37 °C for 15 min, the reaction was stopped and purified by adding equal chloroform volume. Debranched RCA product was precipitated by adding 1/10V 3M sodium acetate (pH 5.2) and 2.5V absolute ethanol followed by incubation at −80 °C for 30 min and centrifugation at 12,000× *g* for 30 min. The obtained pellet was dissolved in nuclease-free water and used for nanopore sequencing.

### 4.5. RT-qPCR Analysis of fl_eccDNA and tr_eccDNAs of ONSEN

To estimate the *ONSEN* eccDNA enrichment profile after RCA, the following primers were designed for the three distinct regions of *ONSEN* sequences: left part—cONS_Lq (TGAAGATCCTAAAGATGGCGAG; TGCTCCTAGGATAGCCTTCA); internal part—cONS_Iq (CTCATGCTCATGTACCGGATG; AGCTTGTAGCCTTTGGAGTTG); right part—cONS_Rq (AAGTCGGCAATAGCTTTGGC; CATACTCCAATTGCACGTCCT). The real-time PCR reaction mix was prepared in 25 μL with the use of 12.5 μL BioMaster HS-qPCR SYBR Blue (2×) (Biolabmix, Russia), 1 μL of 10 pmol of each primer, 9.5 μL of nuclease-free water (Biolabmix, Russia), and 1 μL of RCA product. PCR amplification was run on CFX96 (BioRad, USA) with the following program: 95 °C for 2 min, then 95 °C for 15 s, 58 °C for 15 s, and 40 cycles.

### 4.6. Nanopore Library Preparation and Sequencing

Library preparation was carried out with 1 μg of eccDNA or genomic DNA using the Native Barcoding Expansion 1–12 (Oxford Nanopore Technologies (Oxford, UK), catalog no. EXP-NBD104) and the Ligation Sequencing Kit SQK-LSK109 (Oxford Nanopore Technologies). Sequencing was performed by MinION equipped with an R9.4.1 flow cell. 

### 4.7. Bioinformatic Analysis of eccDNA Sequencing and Data Visualization

The TAIR10 genome was downloaded from the NCBI database. For the analysis of eccDNA, concatemer reads were selected by running TideHunter [28] with the following additional parameters: ‘-f 2 -c 2’ for eccDNA peak determination and ‘-f 2 -c 3’ for eccDNA structure analysis. The selected reads or monomers (for structural analysis) were aligned to the reference genomes TAIR10 using minimap2 software [29] with the following parameters: -ax map-ont -t 100. The obtained sam file was converted to bam format, sorted, and indexed using SAMtools [30]. Then only primary alignments with a minimum MQ value of 30 were left for further analysis using the following command: ‘samtools view -F 3840 -q 30’. The sorted bam files were used for alignment inspection using locally installed JBrowse2 [31]. To detect eccDNA peaks, the genome was split into 500 bp windows using the bedtools [32] command ‘bedtools makewindows -w 500’. The number of reads at each window was determined by ‘bedtools intersect’. To find peaks, the number of reads in each window was compared between control (ZA) and stress variants. The statistical analysis (significance level = 0.01) with two-sided Fisher’s exact test was performed using the Python scipy.stats.fisher_exact function [33] followed by Benjamini–Hochberg correction for multiple comparisons carried out by the function statsmodels.stats.multitest.multipletests (with the argument “method = ‘fdr_bh’”). The obtained results were visualized using the plotnine Python module (https://plotnine.readthedocs.io/, last accessed on 22 February 2023). The code used for this analysis (Jupyter Notebook eccDraw_HS1_vs_mergedZA.ipynb) can be found in the GitHub repository: https://github.com/Kirovez/eccStructONT (last accessed on 27 April 2023).

For the structural analysis of eccDNAs, the Python package ‘eccDNA_struct.py’ (https://github.com/Kirovez/eccStructONT) was written to obtain the matrix that was used for the heatmap visualization performed by ComplexHeatmap R package [34]. This script also generated eccDNA length distribution histograms using seaborn (https://seaborn.pydata.org/, last accessed on 21 April 2023) and pandas (https://pandas.pydata.org/, last accessed on 9 February 2023) modules. 

## 5. Conclusions

In this study, we showed that nanopore sequencing is a useful tool in the detection and structural analysis of eccDNA molecules in plants. Using nanopore sequencing of eccDNA molecules of epigenetically stressed *Arabidopsis* plants grown under various stress treatments (heat, abscisic acid, and flagellin), we showed that TE-derived eccDNA quantities and structures significantly vary between individual TEs. Epigenetic stress alone does not cause eccDNA up-regulation; its combination with heat stress triggers a generation of full-length and various truncated eccDNAs of *ONSEN* elements. We showed that the ratio between full-length and truncated eccDNAs is TE- and condition-dependent. Our work paves the way for further elucidation of eccDNA structural features and their connections with various biological processes, such as eccDNA transcription and eccDNA-mediated TE silencing.

## Figures and Tables

**Figure 1 plants-12-02178-f001:**
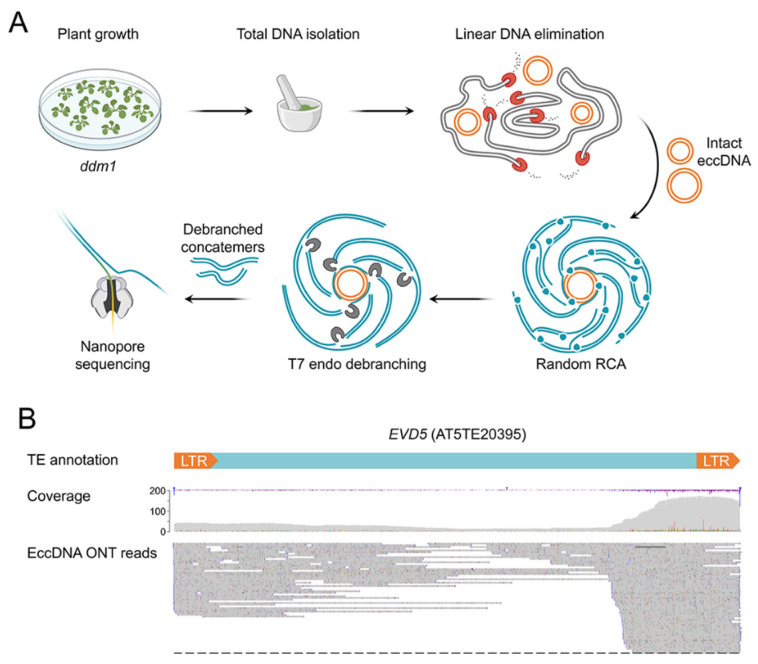
A. thaliana ddm1 mutant eccDNA nanopore sequencing. (**A**) Schematic overview of the eccDNA nanopore sequencing experiment. (**B**) Coverage of EVD5 (AT5TE20395) by eccDNA ddm1 reads.

**Figure 2 plants-12-02178-f002:**
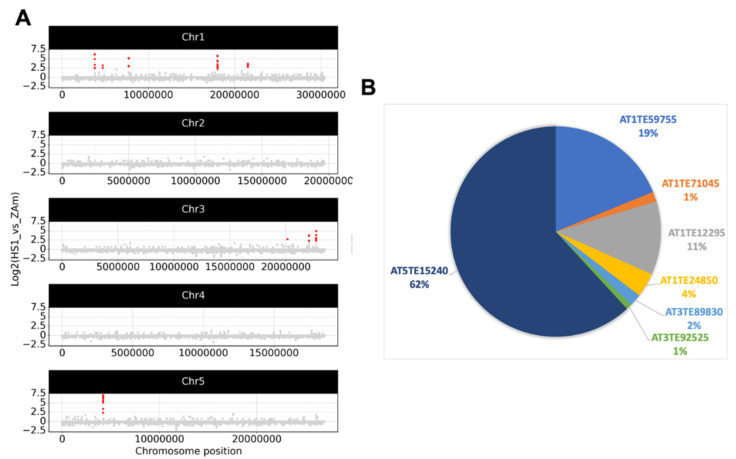
EccDNA loci that were responsive to heat stress. (**A**) Coverage of genomic loci by eccDNA reads in the HS sample. Dots represent the log2 ratio of the number of mapped concatemer reads of eccDNAs for the HS ZA vs. ZA samples (only primary alignments were utilized). In HS ZA samples compared with ZA samples, red dots indicate genomic areas with significantly increased read coverage (Fisher’s exact test with Benjamini–Hochberg correction). (**B**) The ratio of eccDNA reads for seven ONSEN elements is presented as a pie chart.

**Figure 3 plants-12-02178-f003:**
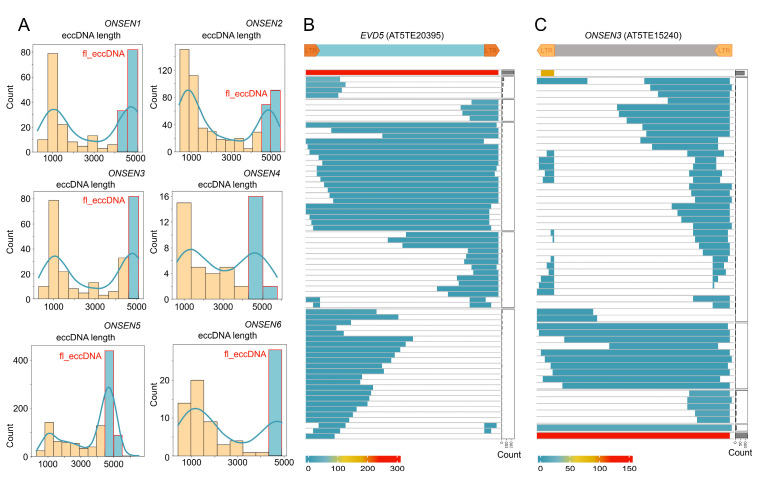
Structure of EVD and ONSEN eccDNAs generated in ddm1 plants under mild heat stress conditions. (**A**) Histograms showing the number of ONSEN eccDNA molecules of different lengths. (**B**,**C**) Heatmaps showing structures and numbers of distinct eccDNAs from EVD and ONSEN3, respectively. The number of eccDNAs in individual groups is illustrated as a bar plot on the right side.

**Figure 4 plants-12-02178-f004:**
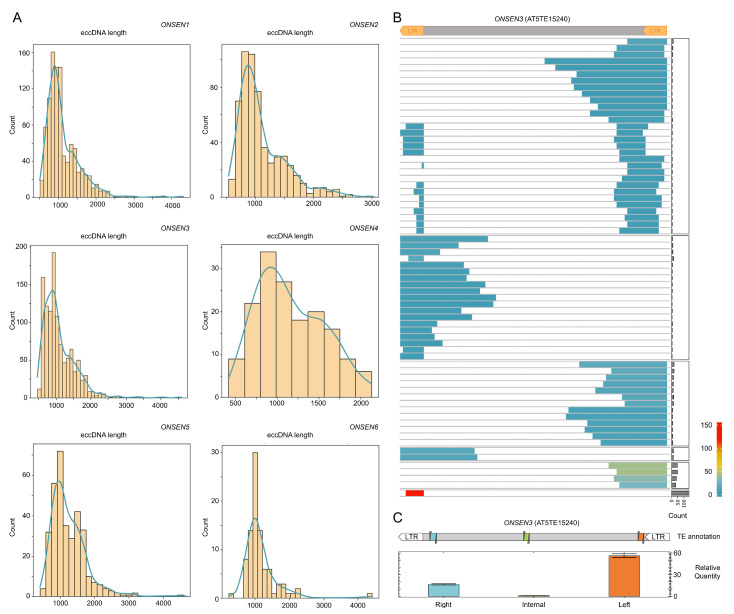
Structures of ONSEN3 eccDNAs detected in Col-0 plants grown in ZA medium and subjected to heat stress conditions. (**A**) Histograms showing the number of ONSEN eccDNA molecules of different lengths. (**B**) Heat maps showing the structures and number of distinct eccDNAs from ONSEN3. The number of eccDNAs in individual groups is illustrated as a bar plot on the right side. (**C**) RT-qPCR results with eccDNA-enriched DNA of ZA HS Col-0 and three primer pairs aligned to the left, right, and central parts of the TE.

## Data Availability

The nanopore data produced for this study are available in the Sequence Read Archive (SRA), NCBI, under Bioproject Accession PRJNA975338.

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
