# Peer review of "Composition and Structure of Arabidopsis thaliana Extrachromosomal Circular DNAs Revealed by Nanopore Sequencing"

_plants, 2023, doi:10.3390/plants12112178_

Round 1
Reviewer 1 Report
Comments:
Throughout the paper the authors need to define acronyms
* Introduction: acronyms need to be defined when first used
- “Col-0 and ddm1 plants' ”
- “RCA”
* Results
· - define "ONT" when first used
· - section 2.2. A positive control is required to indicate that the epigenetic chemicals actually worked
* Discussion
The authors need to provide their version for the entry and exit of TEs from the genome to account for “tr_eccDNAs are byproducts of intra- and inter-molecular recombination events leading to fl_eccDNA conversion into tr_eccDNAs”
Throughout the paper the authors need to define acronyms
Author Response
Dear reviewer,
We appreciate your efforts to improve our manuscript.
Please find our comments below.
Q: Introduction: acronyms need to be defined when first used.
A: We thank the reviewer for the notes. We defined all acronyms in the MS.
Q: section 2.2. A positive control is required to indicate that the epigenetic chemicals actually worked
A: There are no known molecular (e.g. expressed genes or TEs) positive controls for zebularine and α-amanitin treatment. However, it is well-documented fact that these toxins lead to growth and development inhibition in A.thaliana wild type plants (Chun-Hsin Liu, The Plant Cell, 2015; Prochazkova et al., Nucleic Acids Res. 2022). And we clearly observed these growth abnormalities in our experiment where control and ZA plants were compared. To demonstrate this, we added Supplementary Figure S5 displaying the comparison of control and toxin-treated plants development.
Reviewer 2 Report
This study by Merkulov et al. demonstrates the effectiveness of nanopore sequencing in detecting and analyzing eccDNA in plants, specifically Arabidopsis subjected to various stressors. The authors also reveal that a combination of epigenetic stress and heat stress triggers the production of eccDNA, which predominantly appear as full-length or truncated forms. The research further demonstrates that the proportion of these different types of eccDNA is influenced by both the specific TE and the prevailing conditions. Significantly, this work provides additional insights into the nature and behavior of eccDNA, further enriching our understanding of this unique genetic phenomenon. I recommend this manuscript for publication in Plants.
The authors may address these minor concerns.
Page 2: First para: RCA, expand on first instance.
Page 3: Ddm1 to ddm1
Fig 3: The authors mention “we examined ONT data for HS samples and discovered seven eccDNA-producing loci...” Why is the distribution from Onsen7 not shown?
What is the status of Onsen8? Were any eccDNAs detected originating from Onsen8?
4.7 Bioinformatic analysis: “splitted” to “split”
Supplementary Figure S3. ddm1 in italics.
Author Response
We thank the reviewer for the corrections and questions.
The answers are below:
Q: Page 2: First para: RCA, expand on first instance.
A: Corrected.
Q: Page 3: Ddm1 to ddm1
A: Corrected.
Q: Fig 3: The authors mention “we examined ONT data for HS samples and discovered seven eccDNA-producing loci...” Why is the distribution from Onsen7 not shown? What is the status of Onsen8? Were any eccDNAs detected originating from Onsen8?
A: We did not detect the presence of ONSEN8-derived eccDNA in our data. It is in well concordance with previous reports. For example, Roquis et al. (Nucleic Acids Research, 2021) also noted that ONSEN7 copies (ONSEN7 / AT1G21945 / AT1TE24850; ONSEN8 / AT3TE54550 / AT3G32415), have near-zero abundance in eccDNA of A. thaliana exposed to combination of toxin (zebularine; α-amanitin) and heat stress treatments. The previous observations made by Cavrak et al. (PLoS Genetics, 2014) also confirmed the absence of eccDNA derived from both copies in A.thaliana under same conditions. As shown by Ito et al. (Gene, 2012), ONSEN7 and ONSEN8 are the most shared and polymorphic ONSEN copies among natural accessions of A. thaliana pointing to the lost activity due to the age and mutation accumulation.
For ONSEN7, structural analysis was not performed because we did not have sufficient number of eccDNA reads that passed the cutoff used for structural analysis.
Q: 4.7 Bioinformatic analysis: “splitted” to “split”
A: Corrected.
Q: Supplementary Figure S3. ddm1 in italics.
A: Corrected.